# "It's because they're my kids, and I love them": Substance use disorders' impact on children and families: A secondary analysis

Meghan K. Ford[1,2], Ryan Truong[3], Ayesha Shakeel[3], Bruce Knox [2], Susan Bartels[4,5], Colleen Davison[5], Michele Cole[2], Logan Jackson[2], Eva Purkey[2], Imaan Bayoumi [2,5*]

1 Department of Psychology, Queen's University, Humphrey Hall, 62 Arch Street, Kingston, ON, K7L 3N6, Canada, 2 Department of Family Medicine, Queen's University, 220 Bagot Street, Kingston, ON, K7L 5E9, Canada, 3 School of Medicine, Queen's University, 80 Barrie Street, Kingston, ON, K7L 3N6, Canada, 4 Department of Emergency Medicine, Queen's University, 76 Stuart Street, Kingston, ON, K7L 4V7, Canada, 5 Department of Public Health Sciences, Queen's University, 62 Fifth Field Company Lane, Kingston, ON, K7L 3N6, Canada

* bayoumi@queensu.ca

## Abstract

### Background

Substance use disorders (SUD) significantly impact the physical, social, and mental health of individuals, their families, and the wider community. Parental substance use can lead to long-term social and health problems for children. Examining resilience and its determinants among families directly affected by may uncover valuable insights to support families addressing SUD. The existing literature does not adequately address substance use within the context of families with young children and community resilience.

### Aim

The current study aims to enhance our understanding of the daily impact of family members' direct substance use or exposure to indirect substance use within the community on children and families through qualitative interviews.

### Methods

The present study was a qualitative secondary analysis. Families with a self-identified history of adversity and resilience were enrolled in the main study. The qualitative transcripts were analyzed following reflexive thematic analysis.

### Findings

Six families (12 adults, 4 children) were included in the secondary analysis. The analysis generated four themes: (1) How children affect resilience in families affected by SUD; (2) Service needs of parents with SUD to enhance family resilience; (3) The

**Data availability statement:** Data files for this study cannot be shared due to confidentiality. The following documentation, constituting the minimal data set, is available in the Queen's University Dataverse Collection, linked here: https://borealisdata.ca/dataset.xhtml?persistentId=doi:10.5683/SP3/8MH4HF: Qualtrics survey tool, letter of information and consent and assent tools, semi structured focus group interview guides, semi-structured in-depth interview guides, Photovoice interview guides, validation protocol for knowledge dissemination (2024-08-16).

**Funding:** The study was funded by the Social Sciences and Humanities Research Council of Canada (SSHRC). Grant number: 6034834 – received by IB & EP; www.sshrc-crsh.gc.ca. The funders had no role in study design, data collection and analysis, decision to publish, or preparation of the manuscript.

**Competing interests:** The authors have declared that no competing interests exist.

role of social support in family resilience; and (4) How perceptions of safety and trust challenge community resilience.

## Conclusions

The study highlights the significant impact of family and community on the resilience of individuals affected by SUD. It emphasizes the importance of developing addictions services and social environments that are supportive of families with young children and supports the need for services that are substance-free, inclusive, and welcoming to children. Additionally, there is a need to improve service navigation and reduce barriers to care commonly experienced by parents affected by SUD.

## Background

Substance use disorders (SUD) significantly impact the physical, social, and mental health of individuals, their families, and the wider community. Over 20% of Canadians will meet the criteria for having a substance use disorder (SUD) in their lifetime [1]. Substance use disorders are likely to place a substantial burden on individuals, their families, and society [2]. SUD can also have significant impacts on families with young children. Exposure to parental substance use, classified as an adverse childhood experience, can lead to a myriad of short and long-term difficulties for children and adolescents [3,4]. These challenges span various domains, including emotional, behavioural, physical, cognitive, academic, and social aspects [5–7]. For example, parental substance use has been associated with greater attention and conduct difficulties at school, inconsistent attendance, higher chance of school drop out, and poorer academic performance in children [8]. In a study examining long term effects of parental substance use, perceived adult marijuana use was predictive of adolescent substance use [5]. Moreover, parental substance use has been associated with increased risk of child injuries and somatic and psychiatric illness in early childhood [9]. Families with individuals affected by SUD may face social isolation due to stigma, family instability, and financial and relationship difficulties [10]. Far less understood is the process, barriers and facilitators of addressing SUD for families with young children, which may differ from those of individuals without dependents, since these families will also account for the physical, emotional and psychological wellness of their children as well as of themselves.

Deficit-based research can stigmatize and pathologize populations in need [11,12]. Strength-based research supports the identification and promotion of existing resilience within structurally oppressed communities, including individuals and families impacted by SUD [11,12], and leads to the recognition of the importance of individuals' unique strengths and abilities which can be leveraged to overcome adversity. However, strength-based approaches are under-explored in the field of SUD and resilience, particularly where families with young children are concerned [13].

Resilience, commonly defined as the ability of the individual to positively adapt to significant adversity [14], is a dynamic process in which various protective factors

interact to achieve positive outcomes despite hardships such as growing up with parents struggling with SUD or in a community with substantial substance use [14–16]. Family resilience refers to the inherent strengths and adaptive capacities within a family's functioning that enable them to withstand and recover from adversity [17]. Todman and McLaughlin [18] highlight the importance of being aware of and ensuring the presence of protective factors in children's home environments and beyond to address parental substance use and implications for children. This includes considering the child's immediate home environment and family relationships, extended family, social networks and neighbourhood, community resources and service provision, and the broader political system [18]. Examining resilience and its determinants among families directly affected by SUD (e.g., having a parent who misuses substances) or indirectly exposed to substance use (e.g., living in a community impacted by drug use) may uncover valuable insights to support families dealing with SUD.

Further research is needed to better support families impacted by SUD. The current study aims to enhance our understanding of the impact of family member substance use (direct substance use) or exposure to substance use within the community (indirect substance use) on children and families through qualitative interviews and to describe factors supporting family resilience in those affected by substance use.

## Methods

### Study design

The present study was a secondary analysis of a larger main study entitled, "Engaging Families to Build Healthy Communities", which used a multiple case study methodology to explore resilience in the context of the COVID-19 pandemic for families who self identified as resilient and having experienced adversity. The paper reporting on the main study is undergoing peer review. This secondary analysis focused on the ways in which substance use affected families.

### Participants and setting

Recruitment for the main study occurred from January 2022 to March 2023. Families in the Kingston, Frontenac and Lennox and Addington (KFL&A) area were invited to participate in the main study. Participants were recruited with a focus on maximum variation (family composition, ethnocultural). Inclusion criteria included families (1) with at least one adult providing care for a child under the age of 18; (2) with one or more member(s) self-identified as having a history of adversity (e.g., physical, emotional or sexual abuse, poverty, food insecurity, racism, discrimination, poor housing, homelessness); (3) who believed that their family or household member(s) would be interested in participating in a project that created social change, and (4) were able to consent to all components of the study.

Study recruitment was supported through partnering organizations and community members of the CAB, using passive recruitment methods, such as flyers and social media posts, and active recruitment methods such as word of mouth with the people they served.

### Data sources

Data was collected using several research activities: (1) the development of a visual timeline mapping families' experiences during the COVID-19 pandemic and subsequent family group discussion and (2) individual interviews with each family member over twelve years of age, which focused on barriers and facilitators to family resilience. Data for the secondary analysis included all interviews in which participants discussed experiences and perceptions of substance use in data collection activities. In total, the data for 10 families were screened, read, and re-read to determine if there was sufficient data for this study. Transcripts were then explored for discussion about experiences and perceptions of substance use and implications on family resilience. Six families were included in the secondary analysis after the screening process.

Interviews were conducted by two research team members (MF, BK, MC, LJ, EP, IB). The research team was comprised of community based researchers (MC, LJ) with extensive experience working in community based services to

low income families and youth, academic researchers (SB, CD, EP, IB) with equity oriented research programs, project manager (BK) with previous experience in equity based municipal planning and students (PhD candidate MF, and medical students RT, AS) The interviewers did not have pre-existing relationships with research participants. Given the qualitative nature of this study, we acknowledge the importance of positionality and reflexivity in shaping the research process. As researchers, we recognize that our perspectives, lived experiences, and institutional affiliations may have influenced data collection, interpretation, and analysis. To manage potential biases, the research team was comprised of individuals with varying backgrounds in social sciences, medicine, public health, as well as community researchers (community members from various equity-deserving groups with unique lived experiences relevant to the topic of social inclusion) and community engagement. The research team engaged in reflexive practices including reflexive journaling, and well as by collectively reflecting on the data, themes, and results of this study with multiple members considering results to ensure that personal experiences did not unduly bias the results. Additionally, we worked closely with community partners and participants to validate the data, ensuring that knowledge production was collaborative and that participant perspectives remained central to the study. Interviews were completed in-person or virtually based on family preferences. All interviews were audio-recorded and transcribed verbatim, and lasted between one and two hours in length. Field notes were not consistently captured.

### Ethical considerations

Ethics approval was obtained from the Queen's University Health Sciences and Affiliated Teaching Hospital Research Ethics Board (FMED-6810–21; 6034297). Informed, written consent and/or assent were obtained from all study participants, which also included recommendations for community supports for any participants who experienced distress in the research process and for those who identified unmet practical needs. Families received $50.00 CAD for each hour spent associated with data collection. Children aged 12–18 years old who participated in all data collection activities were also entered into a draw for one of three $50.00 CAD gift cards.

Qualitative data were analyzed following a reflexive thematic analysis approach [19] using NVivo 14 (QSR International, 1999) software. Reflexive thematic analysis was chosen due to its flexibility that offered the possibility for an inductively developed analysis that captured both latent and semantic meanings that was informed by critical realism. The theoretical flexibility of reflexive thematic analysis ensured the data analysis captured the lived experiences of children and families impacted by substance use while also locating these experiences within the wider sociocultural context. The six phases of reflexive thematic analysis were undertaken to explore patterns of meaning across the dataset. One author (RT) familiarized themself with the data by reading the interview transcripts, listening to the corresponding audio files, and then coding all transcripts and determined that no new insights had arisen from participant interviews, suggesting data saturation had been achieved. A proportion of the initial coding was completed by two authors (RT and AS) to engage in reflexivity and analysis-enriching discussions to ensure a more comprehensive appreciation of the data. The primary coder (RT) coded the remaining transcripts and engaged with the I-CREAte research team throughout the familiarization, coding, initial theme generation, theme refinement, and writing process to ensure triangulation of the analysis and a thorough understanding of the data [19].

### Results

We included six participating families in this analysis, comprising 12 adults and 4 children. One family that was lost to follow-up did not complete all the research activities, but they did not request the removal of their data, despite being given the option. Therefore, their data were included in this analysis in accordance with the original protocol.

The six families were diverse in terms of family size (two to five members), composition (lone parent female led families, two parent families, and families with biological and fostered children), and child age (infancy to adolescence). Distinctive attributes within individual families included sexual and gender diverse individuals, domestic and sexual abuse

survivors, previously unhoused individuals, individuals with learning and developmental disabilities, recent newcomers, Indigenous peoples, and racialized families. Substance Use Disorder (SUD directly or indirectly impacted all six participating families in their direct family or communities where they lived. In terms of socioeconomic status, families included those on social assistance, financially secure households, individuals with high school level education, and those pursuing or holding a professional degree such as engineering or law.

## Themes

Four themes were generated in the analysis of interviews: 1) How children influence resilience in families affected by SUD; (2) Service needs of parents with SUD to enhance family resilience (3) The role of social support in family resilience; and (4) How perceptions of safety and trust challenge community resilience.

### Theme 1: How children influence resilience in families affected by SUD

Children were described as a source of resilience for families with adults recovering from SUD. Parents described children as anchors to their commitment to sobriety as they expressed a commitment to be present and caring for the wellbeing of their dependents. For example, one mother said:

*"What's keeping us together is our children. That's why I haven't given up on my kids or chosen to go drink a bottle or use up all their money on drugs it's because they are my kids and I love them" (Family 048).*

Several parents in the study noted their children as a source of hope for the future wellbeing of their family. While all parents described aspirations for their children, parents with an experience of substance use were explicit in their desire for their children to break out of the cycles of poverty and substance use that they experienced. One mother shared:

*"I don't want to continue the cycle of abuse, addiction. You know having to live on the system where you stay poor and continue, you know to find other means as you know selling drugs, being …. incarcerated, ending up in treatment facilities"(Family 047).*

Parents reported that their commitment to sobriety was intrinsically tied to their dedication to caring for their children, a commitment that would be compromised if they relapsed or failed to shield their dependents from an environment involving drug use.

### Theme 2: Service needs of parents with SUD to enhance family resilience

While families described children as a source of resilience, parents facing addiction highlighted several challenges in navigating services needed to maintain their sobriety, which they argued hindered their families' overall resilience. For instance, families with young children faced challenges in arranging positive and healthy childcare while undergoing treatments or participating in activities related to addiction recovery, such as attending a 12-step substance use program. Most addictions services are adult-oriented, and parents struggled with balancing their needs with their children's needs including needs related to child routines such as bedtime, homework time, etc or exposure to inappropriate content. When accessing addictions services, one mother shared ongoing moral distress related to her child being exposed to stories of substance use due to lack of access to childcare:

*"It can do more damage than good to drag your kids out to a church basement [at] 7 o'clock at night, which is prime bath, jammie, reading, snuggles time. No. I'm bringing them out in the cold to sit on a chair […] and watch their iPad while everybody talks about how miserable addiction is" (Family 047).*

A lack of access to childcare was an additional stressor for parents seeking sobriety as they strived to use support while keeping their children safe. For example, one father shared the challenges of accessing addiction services without transportation or childcare, necessitating that parents rely on and trust in community services, or even commercial services like taxis, to support them during times of need.

*"The staff [at the methadone clinic] are good, but it's not really a place to really like have my kids around […] that's why every time we go we kind of just take a cab and then we can leave the kids, because we know a lot of the drivers for [taxi company]" (Family 048).*

These challenges also included navigating social programming, including mental health and addiction services. For example, one father noted the importance and value of having access to mental health workers who served as navigators to supportive resources and programming:

*"She helped us by connecting us with supports, helping us get lawyers, help at the table with us, sat down and talked with us for our mental health to figure out what was going wrong, what we needed and how to access it, where to access it, when to access it. Like she went above and beyond her job. Ok like she is what a mental health support worker is supposed to do"(Family 040).*

### Theme 3: The role of social support in family resilience

Parents impacted SUD described a sense of belonging as a critical challenge when maintaining relationships with family, friends, and their local community. Families shared that they struggled to maintain relationships both with people who never struggled with SUD and those who were actively using substances. This sometimes led to profound feelings of social isolation which had important mental health implications for parents, but also for children by limiting their opportunities for socialization. Moreover, families noted the difficulties with maintaining any relationship during the COVID-19 pandemic and associated restrictions. Parents addressing addictions disclosed that people who never struggled with SUD may not understand or respect their sobriety journey. For some adults, this lack of understanding elicited feelings of judgement and lack of support, which stressed their sobriety. For example, a mother shared:

*"The people in my bubble weren't struggling with addiction, so they didn't see an issue with showing up on a random Tuesday with drinks to share while letting our children, […] play together outside." (Family 047)*

Parents remarked that maintaining relationships with people who were currently using substances may challenge their ongoing efforts to be abstinent. Some parents described old friends actively trying to compromise their sobriety. Parents also described the challenges and isolation of not being able to rely on family or friends for childcare if these family members or friends were actively using substances, which directly impacts children. For example, one parent noted not being able to go to their parents' house (their children's grandparents), because they were actively using substances:

*"my parents are still in active addiction. So even going up there, man it's hard sometimes to see and like I don't like seeing them like that, but that's her grandparents, right. So we never go inside. We always just stand in the parking lot and they come out and they'll see them. But even then, like the people that come there and stuff too, it's just not a good environment. That's why we only ever stay for maybe 5 or 10 minutes" (Family 048).*

Maintenance of relationships may be further complicated as adults addressing addiction balance their guilt for abandoning their friends who currently use substances in order to maintain their own sobriety and to protect their children from an

unhealthy environment associated with substance use. For example, one father shared their struggle with finding positive friends:

*"People want to come over and hang out, But it's like – we can't really do it because we don't know half the time if they'll be high when they come over here" (Family 048).*

Despite challenges in finding a positive community of support, families with adults who had these connections repeatedly described how forms of trust and close community were important and helped them feel confident in their recovery. Among the parents in recovery, sponsorship, religious groups, counselling, and supportive family/friends were described as vital to their wellbeing.

**Theme 4: How perceptions of safety and trust challenge community resilience**

Families repeatedly expressed concerns about the impact of community drug use on their feelings of safety and trust in their communities. Parents expressed concerns about the physical risks for their children presented by the presence of people who used substances and evidence of drug use. Specifically, parents worried about used drug paraphernalia and people using substances in community spaces like parks, leading parents to switch parks, or avoid parks:

*Partner: "You could see a park outside our house, which you think would be great for*

*[child]".*

*Mother: "There's needles in it."*

*Partner: "But there's needles and stuff all around it. So we actually take [child] to*

*another park about five minutes away, which is not a problem, of course."*

*Mother: "But annoying"(Family 003)*

Families expressed challenges with building trusting relationships with community members as a consequence of avoiding community spaces in which they perceived drug activity to be occurring. Some families described a lack of trust, and a concern for potential risks in specific community spaces out of fear of harm to themselves or their families. Those who expressed concerns regarding community safety identified concerns regarding implications for children's freedom to play (e.g., to access parks), to engage with neighbours, or to develop their independence (e.g., taking buses, or otherwise being free to move around as older children in environments their parents perceived to be unsafe).

These fears had implications for freedom of movement and independence of youth in some situations. When talking about going downtown, a youth participant shared:

*"I don't think I would be allowed to do that on my own… it's kind of sketchy. I'm sure*

*they'd [parents] let me go with my friends, but I don't think without my friends they'd*

*let me go." (Family 009)*

Families described a variety of perspectives on solutions to substance use and the
    negative sense of community safety. Some families expressed great understanding and empathy for individuals using substances who were unhoused and food insecure due to inflation, while other families suggested an increase in policing might decrease "visible" substance use and therefore improve their sense of community safety. Families proposed solutions such as more employment agencies, a basic income guarantee, and disposal bins for drug paraphernalia in

community spaces. While families expressed perceived challenges with community safety, particularly as it pertains to their perception of safety for their children, and trust in certain spaces, all families expressed a desire to be connected to a supportive community. Specifically, families noted wanting to raise their children in communities where they have trusting relationships with neighbours and the broader community, where basic needs are met, and where families addressing substance use have access to necessary supports to maintain their sobriety. These findings highlight the importance of addressing underlying issues such as affordable housing, mental health resources, poverty, and food insecurity to effectively tackle substance use disorders and enhance perceived community safety.

## Discussion

Our study examined the experiences of direct and indirect exposure to SUD on families and children and identified four important themes including the importance of children as a motivator and source of resilience, the unique service needs of parents with SUD, the importance of social support and the challenges to resilience associated with perceptions of community safety. Consistent with previous research, families with adults with active or historical substance use disorders repeatedly described their children as a source of resilience and motivation in maintaining sobriety. This finding is consistent with literature identifying children and grandchildren as protective factors with respect to sobriety [20].

Although participants described the important role their children played in their sobriety, they also described additional stressors introduced while addressing addictions that interfered with family resilience, such as the need for high quality childcare while accessing SUD services. Institutional barriers such as stigma and discrimination in the delivery of health services, have been well-documented [21,22]. The lack of family centered SUD programming is another important system wide barrier. Many SUD treatment programs are structured for individual adults or couples, and do not account for caregiving responsibilities and children's routines. Moreover, stigma can be even more challenging for caregivers with comorbid substance use and mental health difficulties, and for families with low income who often have limited resources, including housing, food, and employment [22]. This may lead to families being less likely to access important services, either due to fear of stigma, or to unique logistical challenges faced by families with young children. Consistent with our findings, Todman and McLaughlin [18] emphasized that young children in families with substance use are often overlooked. To improve child outcomes, there is a need for policy changes, financial resources, and educational campaigns and training to invest in more family-friendly addiction services [18].

Adults seeking sobriety in our study struggled to access and find social services that were family friendly. Our results underscore the need for system navigation for families trying to navigate the complex system of addiction services. The benefits of system navigation have been highlighted in previous research such as Grycznski et al. (2021), with the authors finding personalized, patient navigation services for SUD patients were effective in reducing hospital readmission [23]. Specifically, having a navigator assist patients in a variety of ways, including talking about substance use, communicating with health care providers, assisting with transportation, arranging appointments, and accompanying patients, successfully reduced barriers to services [23]. Moreover, incorporating inclusive frameworks to care such as the Integrated Strengths-Based Engagement Framework which is composed of four steps (discuss client strengths and establish strength-based goals, select socio-culturally appropriate team members, engage in culturally humble and affirmational care, and measure program satisfaction an self-efficacy outcomes) shows promise in supporting families impacted by SUD [13].

Parents also described how positive community supports have played a critical and beneficial role in the sobriety journey. Many participants with experiences of SUD faced difficulties in maintaining social relationships with friends or family due to misalignment with their sobriety goals, leading to greater isolation. One caregiver reported relapsing due to their desire for child socialization which involved engaging with friends who were socially consuming alcohol and were not aware of the participants's struggles. Our findings are aligned with previous literature suggesting that a lack of social support creates difficulty in sobriety ultimately creating barriers to SUD treatment [24,25]. Social connectedness is an important a protective factor for children living with parental substance use [18].,

Regarding perceptions of community safety and trust, participating families expressed concerns about the risks to their children, which they perceived to be presented by people using substances and substance use paraphernalia in their communities. These concerns were expressed by families with and without personal experience of SUD. Families described a sense of distrust of other community members, and led them to curtail their children's behaviour (e.g., avoiding parks, not allowing youth to walk around the neighbourhood alone), with implications for child well-being, trust, and the development of independence. It is crucial to highlight how the impact of community substance use on individuals' perception of safety can be significantly influenced by stigma. The general population tends to hold significantly more negative views towards individuals with SUD than those with other mental health difficulties [26]. This stigma can exacerbate the divide between community members who do and do not struggle with SUD, reinforcing negative perceptions, and misplaced fears, and potentially contribute to further mistrust and isolation. At a community level, investing in community educational campaigns shows promise in mitigating negative public attitudes towards individuals impacted by SUD [26].

Lastly, families with adults dealing with SUD identified the need for community spaces in which families could avoid contact with substance use paraphernalia and individuals actively using substances. Specifically, long-term investment in improved community infrastructure and outdoor spaces, may improve community comfort and foster community connectedness [27]. Participants also proposed increased social services such as employment agencies and basic income guarantees as interventions that could address some of the root causes of SUD, thereby enhancing individual and family well-being and improving perceived community safety. Individuals who use substances face numerous obstacles to accessing health and social services, including stigma, housing unaffordability, fear of child welfare services, and lack of trust in the healthcare system to a greater extent than those who do not use substances [28,29]. Improving long term outcomes for individuals recovering from SUD and their communities remains contingent upon the wider system capacity to meet their needs [30,31], including integration of mental health and social services (including child welfare services) in collaboration with individuals with SUD and their families.

While this paper provides valuable insight into the unique experiences of families impacted by SUD, there are several limitations to note. First, six families, but only four children were included. More voices from the same community or group that each family represented, including youth are needed. Additionally, the study population did not include any individuals who endorsed actively using substances beyond prescribed opioid replacement therapy during the study. This presents a future area of research to determine if similar themes, such as children as a motivational factor for sobriety, remain consistent or are different among families with members who endorse actively using substances. In addition, participants were asked to recall their experiences over an extended period, potentially affecting their recall. Finally, the inclusion criteria included individuals who were interested in supporting social change, which may not represent the broader views of families experiencing adversity and resilience.

## Conclusions

This study highlights the crucial role of family and community in supporting the sobriety and resilience of individuals affected by SUD. There is a need for family-friendly, strength-based addiction services, system navigation, and hubs to socialize with others seeking sobriety, with an emphasis on creating substance-free and child-friendly environments. Community level engagement, maintenance of environments, and improved infrastructure can improve community perceptions of safety of public community spaces. Policymakers should plan for the distinctive needs of families with children, impacted by SUD to provide safe, accessible addictions/SUD services and alleviate some of the root causes of substance use disorders to support family and community resilience.

## Acknowledgments

The authors would like to acknowledge the project's Community Advisory Board, who helped the authors conceptualize and bring the project to fruition. The Community Advisory Board ensured that this research remained rooted in community needs and followed a rigorous community-based participatory process. Finally, the authors would like to thank the research participants who so willingly shared their personal experiences, who welcomed us into their homes, and trusted us with their stories.

   

## Author contributions

**Conceptualization:** Bruce Knox, Eva Purkey, Logan Jackson, Michele Cole, Susan Bartels, Colleen Davison, Imaan Bayoumi.

**Data curation:** Bruce Knox, Eva Purkey, Susan Bartels, Imaan Bayoumi.

**Formal analysis:** Meghan K Ford, Ayesha Shakeel, Bruce Knox, Eva Purkey, Logan Jackson, Michele Cole, Colleen Davison, Imaan Bayoumi.

**Funding acquisition:** Bruce Knox, Eva Purkey, Colleen Davison, Imaan Bayoumi.

**Investigation:** Meghan K Ford, Bruce Knox, Eva Purkey, Logan Jackson, Michele Cole, Colleen Davison, Imaan Bayoumi.

**Methodology:** Meghan K Ford, Bruce Knox, Eva Purkey, Logan Jackson, Michele Cole, Susan Bartels, Colleen Davison, Imaan Bayoumi.

**Project administration:** Bruce Knox, Eva Purkey, Imaan Bayoumi.

**Resources:** Eva Purkey, Imaan Bayoumi.

**Supervision:** Bruce Knox, Eva Purkey, Colleen Davison, Imaan Bayoumi.

**Validation:** Bruce Knox, Eva Purkey, Imaan Bayoumi.

**Writing – original draft:** Meghan K Ford, Ryan Truong.

**Writing – review & editing:** Meghan K Ford, Ayesha Shakeel, Bruce Knox, Eva Purkey, Logan Jackson, Michele Cole, Susan Bartels, Colleen Davison, Imaan Bayoumi.

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
