## [Decision Letter · Decision Letter 0]

Dear Dr. Bayoumi,

Thank you for submitting your manuscript to PLOS ONE. After careful consideration, we feel that it has merit but does not fully meet PLOS ONE’s publication criteria as it currently stands. Therefore, we invite you to submit a revised version of the manuscript that addresses the points raised during the review process.

We look forward to receiving your revised manuscript.

Kind regards,

Anthony A. Olashore, MBCHB, PhD.

Academic Editor

PLOS ONE

Journal Requirements:

4. Your abstract cannot contain citations. Please only include citations in the body text of the manuscript, and ensure that they remain in ascending numerical order on first mention.

5. Please remove all personal information, ensure that the data shared are in accordance with participant consent, and re-upload a fully anonymized data set.

Reviewers' comments:

Reviewer's Responses to Questions

**Comments to the Author**

1. Is the manuscript technically sound, and do the data support the conclusions?

Reviewer #1: Partly

Reviewer #2: Yes

Reviewer #3: Yes

Reviewer #4: Yes

Reviewer #5: Yes

2. Has the statistical analysis been performed appropriately and rigorously?

Reviewer #1: N/A

Reviewer #2: Yes

Reviewer #3: N/A

Reviewer #4: Yes

Reviewer #5: Yes

3. Have the authors made all data underlying the findings in their manuscript fully available?

Reviewer #1: No

Reviewer #2: Yes

Reviewer #3: No

Reviewer #4: Yes

Reviewer #5: Yes

4. Is the manuscript presented in an intelligible fashion and written in standard English?

Reviewer #1: No

Reviewer #2: Yes

Reviewer #3: Yes

Reviewer #4: Yes

Reviewer #5: Yes

Reviewer #1: This is an important study addressing an important global health issue.

Authors need to confine themselves to reporting of secondary data analysis as well as explain the conceptual framework they use to guide that secondary analysis.

Additionally, authors need to subject their manuscript to language editing service.

Reviewer #2: 1. would have loved to see the raw data as it emerges from the analysis, but its still ok as the authors elaborated on how they conducted the analysis.

2. Research on adversity can often rekindle past unpleasant memories, authors should always put up a mechanisms for those who may feel unsettled with such interviews. Authors did not mention or put up anything should that have occured

Reviewer #3: This is a good paper, but I believe it needs to be reviewed for a better presentation.

This is a secondary study, but the way it is presented suggests that data was obtained from participants, despite the fact that it was extracted from the primary data source.

Methods:

Line 101 to 144: "...study entitled, “Engaging Families to Build Healthy Communities" Is this study already been published? If yes, I suggest that authors reference the paper and delete sentences that describe the primary study. I tried to get the primary study; so far I could only access the abstract (https://www.annfammed.org/content/22/Supplement_1/6251). If it was not published, the authors should provide a summary of the primary study and concentrate on the method by which they extracted data from the primary source: sampling strategies and procedure to include data, as it from how many participants? ...

Line 145: Ethical Considerations: This should be more about manipulation of data from the primary source; was it done in an ethical manner?

Also, techniques to enhance trustworthiness and credibility of data analysis (member checking, audit trail, triangulation).

Results:

Authors should remember that in the secondary study they were dealing with data from participants and not with participants directly. So this should be stated as (line 181): Data from six families were extracted from the primary study for analysis in this study.

Reviewer #4: This is well written paper that explores a relatively neglected angle in addiction care. Despite the small sample the insights gleaned can add to improving care. Just one question: could the fact that inclusion criteria included families who believed that their family or household members were interested in participating in a project that "created social change" have led to a bias towards recruiting participants with a strong desire for social change and therefore inadvertently excluded those withut such desire?. Perhaps note could be taken of this under limitations.

Reviewer #5: The manuscript is technically sound, and the data supports the conclusions. The analyses was performed as expected and the manuscript is written in clear intelligible English.

The manuscript is acceptable for publishing as presented.

**Do you want your identity to be public for this peer review?** For information about this choice, including consent withdrawal, please see our Privacy Policy

Reviewer #1: No

Reviewer #2: **Yes: ** Hlanganiso Roy

Reviewer #3: **Yes: ** Stephane Tshitenge

Reviewer #4: No

Reviewer #5: **Yes: ** Olorunfemi Oladotun Ogunwobi

---

## [Author Response · Author response to Decision Letter 1]

14 Apr 2025

Response to reviewers

Journal requirements

1. Please ensure that your manuscript meets PLOS ONE's style requirements, including those for file naming. The title has been edited as recommended.

2. Your ethics statement should only appear in the Methods section of your manuscript. If your ethics statement is written in any section besides the Methods, please delete it from any other section. The ethics statement was removed from the Acknowledgements section, since it already appeared in Methods.

3. When completing the data availability statement of the submission form, you indicated that you will make your data available on acceptance. We strongly recommend all authors decide on a data sharing plan before acceptance, as the process can be lengthy and hold up publication timelines. Please note that, though access restrictions are acceptable now, your entire data will need to be made freely accessible if your manuscript is accepted for publication. This policy applies to all data except where public deposition would breach compliance with the protocol approved by your research ethics board. If you are unable to adhere to our open data policy, please kindly revise your statement to explain your reasoning and we will seek the editor's input on an exemption. Please be assured that, once you have provided your new statement, the assessment of your exemption will not hold up the peer review process. Data files for this study cannot be shared due to confidentiality. The following documentation is available: Qualtrics survey tool, letter of information and consent and assent tools, semi structured focus group interview guides, semi-structured in-depth interview guides, Photovoice interview guides, validation protocol for knowledge dissemination (2024-08-16). The Data Availability Statement has been updated.

4. Your abstract cannot contain citations. Please only include citations in the body text of the manuscript, and ensure that they remain in ascending numerical order on first mention. There are no citations in the abstract.

5. Please remove all personal information, ensure that the data shared are in accordance with participant consent, and re-upload a fully anonymized data set. All information shared is in accordance with participant consent. No personal information is included.

6. Please include captions for your Supporting Information files at the end of your manuscript, and update any in-text citations to match accordingly.

7. Please review your reference list to ensure that it is complete and correct. If you have cited papers that have been retracted, please include the rationale for doing so in the manuscript text, or remove these references and replace them with relevant current references. Any changes to the reference list should be mentioned in the rebuttal letter that accompanies your revised manuscript. If you need to cite a retracted article, indicate the article’s retracted status in the References list and also include a citation and full reference for the retraction notice. References have been corrected. Specifically, the full reference #1 has been added and updated and missing information was added to reference #18

Reviewer 1

This is an important study addressing an important global health issue.

Authors need to confine themselves to reporting of secondary data analysis as well as explain the conceptual framework they use to guide that secondary analysis.

Additionally, authors need to subject their manuscript to language editing service. The manuscript has been edited for clarity and is focused on the secondary analysis. The main study is now only briefly discussed.

Reviewer 2

1. would have loved to see the raw data as it emerges from the analysis, but its still ok as the authors elaborated on how they conducted the analysis.

We are unable to share the raw data due to confidentiality but have provided additional detail in

2. Research on adversity can often rekindle past unpleasant memories, authors should always put up a mechanisms for those who may feel unsettled with such interviews. Authors did not mention or put up anything should that have occured Thank you for pointing out this important need. We have updated the ethics section to describe the process of recommending available community supports for participants who needed them.

Reviewer 3

This is a good paper, but I believe it needs to be reviewed for a better presentation.

This is a secondary study, but the way it is presented suggests that data was obtained from participants, despite the fact that it was extracted from the primary data source.

Methods:

Line 101 to 144: "...study entitled, “Engaging Families to Build Healthy Communities" Is this study already been published? If yes, I suggest that authors reference the paper and delete sentences that describe the primary study. I tried to get the primary study; so far I could only access the abstract (https://www.annfammed.org/content/22/Supplement_1/6251). If it was not published, the authors should provide a summary of the primary study and concentrate on the method by which they extracted data from the primary source: sampling strategies and procedure to include data, as it from how many participants? ...

Line 145: Ethical Considerations: This should be more about manipulation of data from the primary source; was it done in an ethical manner?

Also, techniques to enhance trustworthiness and credibility of data analysis (member checking, audit trail, triangulation).

Results:

Authors should remember that in the secondary study they were dealing with data from participants and not with participants directly. So this should be stated as (line 181): Data from six families were extracted from the primary study for analysis in this study.

Thank you for the comment. Reporting on the primary study is currently under review, which was clarified in the text. We have described in additional detail the validation of results with participants and with community partners and triangulation of analysis.

We think Ethical consideration should related to ethics in the conduct of research. We have summarized the ethical considerations and hope it is clear.

Reviewer 4

This is well written paper that explores a relatively neglected angle in addiction care. Despite the small sample the insights gleaned can add to improving care. Just one question: could the fact that inclusion criteria included families who believed that their family or household members were interested in participating in a project that "created social change" have led to a bias towards recruiting participants with a strong desire for social change and therefore inadvertently excluded those withut such desire?. Perhaps note could be taken of this under limitations. Thank you – we have added this observation to the study limitations.

Reviewer 5

The manuscript is technically sound, and the data supports the conclusions. The analyses was performed as expected and the manuscript is written in clear intelligible English.

The manuscript is acceptable for publishing as presented. Thank you.

---

## [Decision Letter · Decision Letter 1]

Dear Dr. Bayoumi,

Thank you for submitting your manuscript to PLOS ONE. After careful consideration, we feel that it has merit but does not fully meet PLOS ONE’s publication criteria as it currently stands. Therefore, we invite you to submit a revised version of the manuscript that addresses the points raised during the review process.

We look forward to receiving your revised manuscript.

Kind regards,

Anthony A. Olashore, MBCHB, PhD, FWACP

Academic Editor

PLOS ONE

Journal Requirements:

Reviewers' comments:

Reviewer's Responses to Questions

**Comments to the Author**

Reviewer #1: All comments have been addressed

Reviewer #2: All comments have been addressed

Reviewer #3: All comments have been addressed

2. Is the manuscript technically sound, and do the data support the conclusions?

Reviewer #1: Yes

Reviewer #2: Yes

Reviewer #3: Yes

3. Has the statistical analysis been performed appropriately and rigorously?

Reviewer #1: Yes

Reviewer #2: Yes

Reviewer #3: N/A

4. Have the authors made all data underlying the findings in their manuscript fully available?

Reviewer #1: Yes

Reviewer #2: Yes

Reviewer #3: Yes

5. Is the manuscript presented in an intelligible fashion and written in standard English?

Reviewer #1: Yes

Reviewer #2: Yes

Reviewer #3: Yes

Reviewer #1: The authors are commended for satisfactorily responding to the raised comments. This is an important study that adds to the body of knowledge regarding this important issue.

Reviewer #2: My earlier concerns have been attended to, especially on ethics which are key in any human subject research studies

Reviewer #3: I understand that this paper uses data from the main study “Engaging Families to Build Healthy Communities” that is yet to be published. This is a secondary analysis aimed at exploring new questions or applying different analytical methods to gain additional insights (specifically, the ways in which substance use affected families). To avoid confusion, I suggest that the authors replace terms like "primary study" in lines 90 and 102... with "main study," as mentioned in line 98.

The paper meets the trustworthiness criteria (credibility, transferability, dependability, and confirmability); however, the authors need to mention how saturation of data was achieved, duration of the interviews or group discussions, whether field notes were taken during interviews or group discussions. Few sentences can be added to meet the 32-item checklist COREQ (Consolidated criteria for reporting qualitative studies).

**Do you want your identity to be public for this peer review?** For information about this choice, including consent withdrawal, please see our Privacy Policy

Reviewer #1: No

Reviewer #2: **Yes: ** Hlanganiso Roy

Reviewer #3: **Yes: ** stephane tshitenge

---

## [Author Response · Author response to Decision Letter 2]

13 Jun 2025

Reviewer Feedback Author Response

1. If the authors have adequately addressed your comments raised in a previous round of review and you feel that this manuscript is now acceptable for publication, you may indicate that here to bypass the “Comments to the Author” section, enter your conflict of interest statement in the “Confidential to Editor” section, and submit your "Accept" recommendation.

Reviewer #1: All comments have been addressed

Reviewer #2: All comments have been addressed

Reviewer #3: All comments have been addressed Thank you.

Is the manuscript technically sound, and do the data support the conclusions?

Reviewer #1: Yes

Reviewer #2: Yes

Reviewer #3: Yes Thank you.

Has the statistical analysis been performed appropriately and rigorously?

Reviewer #1: Yes

Reviewer #2: Yes

Reviewer #3: N/A

Thank you.

Have the authors made all data underlying the findings in their manuscript fully available?

Reviewer #1: Yes

Reviewer #2: Yes

Reviewer #3: Yes Thank you.

Is the manuscript presented in an intelligible fashion and written in standard English?

Reviewer #1: Yes

Reviewer #2: Yes

Reviewer #3: Yes Thank you.

Review Comments to the Author

Reviewer #1: The authors are commended for satisfactorily responding to the raised comments. This is an important study that adds to the body of knowledge regarding this important issue.

Thank you.

Reviewer #2: My earlier concerns have been attended to, especially on ethics which are key in any human subject research studies Thank you.

Reviewer #3: I understand that this paper uses data from the main study “Engaging Families to Build Healthy Communities” that is yet to be published. This is a secondary analysis aimed at exploring new questions or applying different analytical methods to gain additional insights (specifically, the ways in which substance use affected families). To avoid confusion, I suggest that the authors replace terms like "primary study" in lines 90 and 102... with "main study," as mentioned in line 98.

We have changed the term ‘primary study’ to ‘main study’ throughout the manuscript

The paper meets the trustworthiness criteria (credibility, transferability, dependability, and confirmability); however, the authors need to mention how saturation of data was achieved, duration of the interviews or group discussions, whether field notes were taken during interviews or group discussions. Few sentences can be added to meet the 32-item checklist COREQ (Consolidated criteria for reporting qualitative studies). Thank you for this helpful feedback.

We note in line 144/145 that ‘All interviews were audio-recorded and transcribed verbatim, and lasted between one and two hours in length.’

We have added a note indicating that field notes were not consistently captured.

We also added the methods for determining data saturation on line 163/164.

---

## [Decision Letter · Decision Letter 2]

“It’s because they’re my kids, and I love them”: Substance Use Disorders’ Impact on Children and Families: A Secondary Analysis.

PONE-D-24-36212R2

Dear Dr. Bayoumi,

We’re pleased to inform you that your manuscript has been judged scientifically suitable for publication and will be formally accepted for publication once it meets all outstanding technical requirements.

Kind regards,

Anthony A. Olashore, PhD.

Academic Editor

PLOS ONE

**Comments to the Author**

Reviewer #3: All comments have been addressed

2. Is the manuscript technically sound, and do the data support the conclusions?

Reviewer #3: Yes

3. Has the statistical analysis been performed appropriately and rigorously?

Reviewer #3: N/A

4. Have the authors made all data underlying the findings in their manuscript fully available?

Reviewer #3: Yes

5. Is the manuscript presented in an intelligible fashion and written in standard English?

Reviewer #3: Yes

Reviewer #3: All of my earlier remarks have been addressed. The paper meet the 32-item checklist COREQ (Consolidated criteria for reporting qualitative studies).

**Do you want your identity to be public for this peer review?** For information about this choice, including consent withdrawal, please see our Privacy Policy

Reviewer #3: No

---

## [Editor Report · Acceptance letter]

PONE-D-24-36212R2

PLOS ONE

Dear Dr. Bayoumi,

I'm pleased to inform you that your manuscript has been deemed suitable for publication in PLOS ONE. Congratulations! Your manuscript is now being handed over to our production team.

Kind regards,

on behalf of

Dr. Anthony A. Olashore

Academic Editor

PLOS ONE